# Quantum Control of a Nonlinear Time-Dependent Interaction of a Damped Three-Level Atom

Sameh Korashy [1,*] and Mahmoud Abdel-Aty [1,2]

1   Department of Mathematics, Faculty of Science, Sohag University, Sohag 82524, Egypt; abdelatyquantum@gmail.com
2   Deanship of Graduate Studies and Research, Ahlia University, Manama 10878, Bahrain
*   Correspondence: skorashe@yahoo.com

**Abstract:** We investigate some new aspects of the nonlinear interaction between a three-level $\Xi$-type atom and bimodal field. The photon-assisted atomic phase damping, detuning parameter, Kerr nonlinearity and the time-dependent coupling have been considered. The general solution has been obtained by using the Schrödinger equation when the atom and the field are initially prepared in the excited state and coherent state, respectively. The atomic population inversion and concurrence are discussed. It is shown that the time-dependent coupling parameter and the detuning parameter can be considered as quantum controller parameters of the atomic population inversion and quantum entanglement in the considered model.

**Keywords:** three-level atom; time-dependent coupling parameter; Kerr medium; atomic population inversion; concurrence

## 1. Introduction

One of the most famous models in quantum optics is the Jaynes–Cummings model (JCM) [1], which includes a single two-level atom interacting with a single near-resonant quantized cavity mode of the electromagnetic field. It plays a fundamental rule in quantum optics due to the experimental realization of the nonclassical effects. It is experimentally realizable and has undergone many theoretical studies [2].

Several modifications and generalizations have been made to the JCM in many different directions. For example, multi-photon transition, multi-level atoms, intensity-dependent coupling, multi-atoms interaction, multi-mode fields, Stark shift and Kerr nonlinearity have been studied in recent decades [3–13].

A lot of researches are focused on studying multi-level atomic systems in different areas of quantum optics. One of the interesting examples of the generalized JCM is a system of three-level atom different configurations ($\Lambda$, $V$, and $\Xi$) and one- or two-mode fields [3,4,14,15]. Many studies has been conducted on the atom-field entanglement and geometric phase in such systems [3,4,13,16]. A lot of studies have investigated a three-level atom in motion which interacts with a single-mode field in an optical cavity in an intensity-dependent coupling regime [17]. Dynamics of entropy and nonclassical properties of the state of a $\Lambda$-type three-level atom interacting with a single-mode cavity field with intensity-dependent coupling in a Kerr medium have been studied in [18].

The damping is a well-known phenomenon in quantum information processing. Several papers have studied the damping effects on entanglement and some nonclassical properties. Several studies have investigated the phase damping in the JCM [19–21] and its influence in quantum properties of the multi-quanta two-mode JCM [22]. The time-dependent interaction between a three-level $\Lambda$-type atom two-mode electromagnetic field in a Kerr-like medium, where the field and the atom are suffering decay rates, has been

studied [23]. In addition, the time-dependent interaction between two Λ-type three-level atoms and a single-mode cavity field has been discussed [24] when the damping parameter is taken into account. In recent years, much attention has been focused on the properties of multi-atoms and multi-level atomic systems when time-dependent coupling with the field is considered [23–29]. More recently, the entanglement and entropy squeezing for moving two two-level atoms interacting with a radiation field have been investigated in [30].

In this paper, we extend the investigations in [31] to study the dynamics of a three-level Ξ-type atom interacting with a two-mode coherent field. Furthermore, the field and the atom are assumed to be coupled with modulated coupling parameter, which depends explicitly on time. In order to discuss the dynamics of the present system, we will find the solution of the wave function in the Schrődinger picture under certain approximation similar to that of the Rotating-Wave Approximation (RWA) at any time $t > 0$. This is performed in the next section where we also derive the reduced density matrix of the atom. In Section 3, we employ our results to calculate the atomic population inversion and the dynamical properties for different regimes. We devote Section 4 to the discussion of the degree of entanglement, where we use the definition of concurrence. Finally, we give our conclusions in Section 5.

## 2. Physical Model

The considered model is a time-dependent regime that consists of a moving three-level (Ξ-type) atom with the energy levels $\omega_1 > \omega_2 > \omega_3$, which interacts with a two-mode field of frequency $\Omega_j$ in an optical cavity surrounded by Kerr nonlinearity in the presence of detuning parameters. The transitions $|1\rangle \longleftrightarrow |2\rangle$ and $|2\rangle \longleftrightarrow |3\rangle$ are allowed, while the transition $|1\rangle \longleftrightarrow |3\rangle$ is forbidden, as shown in Figure 1. The interaction Hamiltonian in the Rotating-Wave Approximation (RWA) of the introduced physical system [31,32] ($\hbar = 1$):

$$
\begin{aligned}
\hat{H}_I = {} & f_1(t)(\hat{a}e^{i\Delta_1 t}\hat{\sigma}_{12} + \hat{a}^\dagger e^{-i\Delta_1 t}\hat{\sigma}_{21}) + f_2(t)(\hat{b}\,e^{-i\Delta_2 t}\hat{\sigma}_{23} + \hat{b}^\dagger\,e^{i\Delta_2 t}\hat{\sigma}_{32}) \\
& + \chi_1\hat{a}^{\dagger 2}\hat{a}^2 + \chi_2\,b^{\dagger 2}\,\hat{b}^2 - \frac{i}{2}\gamma_1\hat{n}_1(\hat{\sigma}_{11} + \hat{\sigma}_{22}) - \frac{i}{2}\gamma_2\hat{n}_2(\hat{\sigma}_{22} + \hat{\sigma}_{33}),
\end{aligned} \tag{1}
$$

Here, the operator $\hat{\sigma}_{ij} = |i\rangle\langle j|$ is the atomic raising or lowering operator, the operators $\hat{a}^\dagger$, $\hat{b}^\dagger$ are the field creation operators of the field mode, and the operators $\hat{a}$, $\hat{b}$ are the field annihilation operators of the field mode. Meanwhile, $f_i(t)$, $i = 1, 2$, are the atom-field coupling parameters, $\chi_j$ is the third-order nonlinearity of the Kerr medium, and $\gamma_i$, $i = 1, 2$, represent the photon-assisted atomic phase damping parameters, which are positive and real. The detuning parameters $\Delta_1$, $\Delta_2$ are given by

$$
\begin{aligned}
\Delta_1 &= \omega_1 - \omega_2 - \Omega_1, \\
\Delta_2 &= \Omega_2 - (\omega_2 - \omega_3),
\end{aligned} \tag{2}
$$

$\Omega_j$, $j = 1, 2$ is the frequency of the field mode. We consider $f_1(t) = f_2(t) = f(t) = \lambda_j \cos(\mu t) = \frac{\lambda_j}{2}(e^{i\mu t} + e^{-i\mu t})$, where, $\lambda_j$, $\mu$, $j = 1, 2$ are arbitrary constants. As one can see, there are two exponential terms in the Hamiltonian: one contains rapidly oscillating terms $e^{\pm i(\Delta_j + \mu)t}$, and the other contains slowly varying terms $e^{\pm i(\Delta_j - \mu)t}$. In this case, if we neglect the rapidly varying terms compared with the slowly varying terms, then the interaction Hamiltonian can be rewritten in the following manner

$$
\begin{aligned}
\hat{H}_I = {} & \frac{\lambda_1}{2}(\hat{a}e^{i\delta_1 t}\hat{\sigma}_{12} + \hat{a}^\dagger e^{-i\delta_1 t}\hat{\sigma}_{21}) + \frac{\lambda_2}{2}(\hat{b}e^{-i\delta_2 t}\hat{\sigma}_{23} + \hat{b}^\dagger e^{i\delta_2 t}\hat{\sigma}_{32}) \\
& + \chi_1\hat{a}^{\dagger 2}\hat{a}^2 + \chi_2\hat{b}^{\dagger 2}\hat{b}^2 - \frac{i}{2}\gamma_1\hat{n}_1(\hat{\sigma}_{11} + \hat{\sigma}_{22}) - \frac{i}{2}\gamma_2\hat{n}_2(\hat{\sigma}_{22} + \hat{\sigma}_{33})
\end{aligned} \tag{3}
$$

where

$$
\delta_1 = \Delta_1 - \mu, \ \delta_2 = \Delta_2 - \mu. \tag{4}
$$

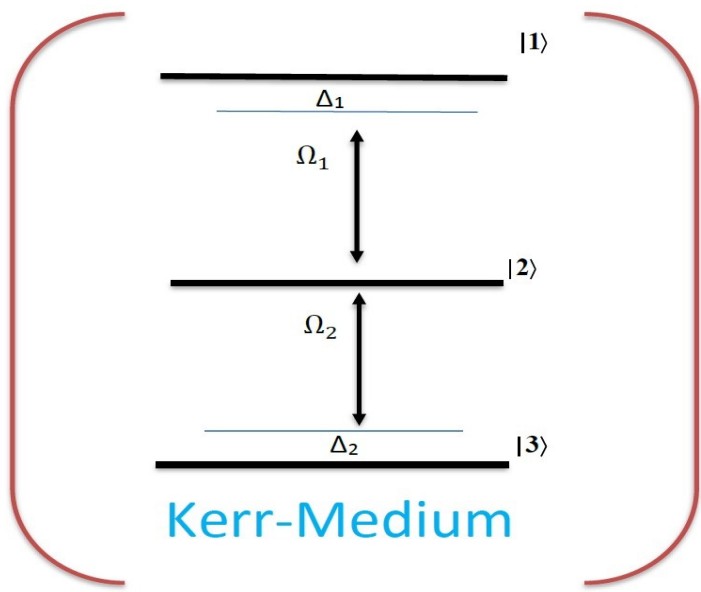

**Figure 1.** Schematic diagram of a three-level $\Xi$-type atom interacting with a two-mode field.

We assume that the wave function of the atom field at any time $t > 0$ can be expressed as

$$
|\Psi(t)\rangle = \sum_{n_1, n_2 = 0}^{\infty} [G_1(n_1, n_2, t)|1, n_1, n_2\rangle + G_2(n_1 + 1, n_2, t)|2, n_1 + 1, n_2\rangle \tag{5}
$$
$$
+ G_3(n_1 + 1, n_2 + 1, t)|3, n_1 + 1, n_2 + 1\rangle]
$$

To reach this goal, suppose that the atom-field initial state is

$$
|\Psi(0)\rangle = \sum_{n_1, n_2 = 0}^{\infty} q_{n_1} q_{n_2} |1, n_1, n_2\rangle \tag{6}
$$

where $q_{n_i} = e^{-|\alpha_i|^2/2} \frac{\alpha_i^{n_i}}{\sqrt{n_i!}}$, $|\alpha_i|^2 = \bar{n}_i$ is the initial mean photon number for the mode. Now, by substituting $|\Psi(t)\rangle$ from Equation (5) and $\hat{H}_I$ from Equation (3) in the time-dependent Schrödinger equation $i\frac{\partial}{\partial t}|\Psi(t)\rangle = \hat{H}_I |\Psi(t)\rangle$, one may arrive at the following coupled differential equations for the atomic probability amplitudes

$$
i\frac{d}{dt}\begin{pmatrix} G_1 \\ G_2 \\ G_3 \end{pmatrix} = \begin{pmatrix} \bar{\alpha}_1 & v_1 e^{i\delta_1 t} & 0 \\ v_1 e^{-i\delta_1 t} & \bar{\alpha}_2 & v_2 e^{-i\delta_2 t} \\ 0 & v_2 e^{i\delta_2 t} & \bar{\alpha}_3 \end{pmatrix}\begin{pmatrix} G_1 \\ G_2 \\ G_3 \end{pmatrix}. \tag{7}
$$

where

$$
\begin{aligned}
\bar{\alpha}_1 &= \chi_1(n_1)(n_1 - 1) + \chi_2(n_2)[(n_2 - 1), \\
\bar{\alpha}_2 &= \chi_1(n_1)(n_1 + 1) + \chi_2(n_2)(n_2 - 1), \\
\bar{\alpha}_3 &= \chi_1(n_1)(n_1 + 1) + \chi_2(n_2)(n_2 + 1), \\
v_1 &= \frac{\lambda_1}{2}\sqrt{n_1 + 1}, \; v_2 = \frac{\lambda_2}{2}\sqrt{n_2 + 1}.
\end{aligned} \tag{8}
$$

The solution of Equation (7) is given as follows

$$
\begin{aligned}
G_1 &= \sum_{j=1}^{3} B_j e^{i\xi_j t}, \\
G_2 &= -\frac{1}{v_1} \sum_{j=1}^{3} B_j (\bar{\alpha}_1 + \xi_j) e^{i(\xi_j - \delta_1)t}, \\
G_3 &= \frac{1}{v_1 v_2} \sum_{j=1}^{3} B_j [(\xi_j + \bar{\alpha}_2 - \delta_1)(\bar{\alpha}_2 + \xi_j) - v_1^2] e^{i(\xi_j - \delta_1 + \delta_2)t},
\end{aligned}
\tag{9}
$$

By applying these initial conditions for the atom and field and using (9), the $B_j$ coefficients read as

$$
B_j = \frac{[(\Gamma_3 + \xi_k + \xi_l)\bar{\alpha}_1 + \xi_k \xi_l - \Gamma_4] q_{n_1} q_{n_2}}{\xi_{jk}\, \xi_{jl}}, \quad j \neq k = 1,\ 2,
\tag{10}
$$

where $\xi_{jk} = \xi_j - \xi_k$, $\xi_j$, $j = 1$, 2, are the roots of the following third-order algebraic equation

$$
\xi^3 + h_1 \xi^2 + h_2 \xi + h_3 = 0,
\tag{11}
$$

where

$$
\begin{aligned}
h_1 &= \Gamma_1 + \Gamma_3 + \bar{\alpha}_3, \quad h_2 = \Gamma_1 \Gamma_3 + \Gamma_4 + \bar{\alpha}_3 \Gamma_3 - V_2^2, \\
h_3 &= \Gamma_1 \Gamma_4 + \bar{\alpha}_3 \Gamma_4 - \bar{\alpha}_1 V_2^2, \quad \Gamma_1 = \delta_2 - \delta_1, \\
\Gamma_2 &= \bar{\alpha}_2 - \delta_1, \ \Gamma_3 = \delta_1 + \Gamma_2, \ \Gamma_4 = \bar{\alpha}_1 \Gamma_2 - V_1^2.
\end{aligned}
\tag{12}
$$

The three roots of the third-order Equation (11) are given in the following form [33]

$$
\begin{aligned}
\xi_m &= -\frac{1}{3}h_1 + \frac{2}{3}\sqrt{h_1^2 - 3h_2}\, \cos(\Phi + \frac{2}{3}(m-1)\pi), \ m = 1,2,3, \\
\Phi &= \frac{1}{3}\arccos\left[\frac{9h_1 h_2 - 2h_1^3 - 27h_3}{2(h_1^2 - 3h_2)^{2/3}}\right].
\end{aligned}
\tag{13}
$$

At any time $t > 0$, the reduced density matrix of the atom describing the system is given by:

$$
\hat{\varrho}(t) = \begin{pmatrix} \varrho_{11}(t) & \varrho_{12}(t) & \varrho_{13}(t) \\ \varrho_{21}(t) & \varrho_{22}(t) & \varrho_{23}(t) \\ \varrho_{31}(t) & \varrho_{32}(t) & \varrho_{33}(t) \end{pmatrix},
\tag{14}
$$

where

$$
\begin{aligned}
\varrho_{11}(t) &= \sum_{n_1,n_2=0}^{\infty} G_1(n_1, n_2, t) G_1^*(n_1, n_2, t), \\
\varrho_{22}(t) &= \sum_{n_1,n_2=0}^{\infty} G_2(n_1 + 1, n_2, t) G_2^*(n_1 + 1, n_2, t), \\
\varrho_{33}(t) &= \sum_{n_1,n_2=0}^{\infty} G_3(n_1 + 1, n_2 + 1, t) G_3^*(n_1 + 1, n_2 + 1, t), ..., \\
\varrho_{il}(t) &= \varrho_{li}^*(t).
\end{aligned}
\tag{15}
$$

In the next sections, for simplisity, we consider the constants $\lambda_i = \lambda$ have been taken to be real, and the interaction time is the scaled time $\tau = \lambda t$.

### 3. Atomic Population Inversion

In fact, we can obtain information about the behavior of the atom–field interaction through the collapse and revival phenomenon. So, we shall study the dynamics of an important quantity, namely atomic population inversion. The atomic inversion is defined as the difference between the exited state $|1\rangle$ and the ground state $|3\rangle$ which may be written as follows [3]

$$W(t) = \varrho_{11}(t) - \varrho_{33}(t). \tag{16}$$

Now, we shall study the behavior of the atomic population inversion in the time-dependent case for $\bar{n}_1 = \bar{n}_2 = 10$ . This will be completed on the basis of the previous calculations. We examine the influence of the time-dependent coupling parameter, detuning parameters, and Kerr medium on the behavior of the atomic population inversion in the absence or presence of the photon-assisted atomic phase damping parameter. The temporal evolution atomic population inversion has been given in Figures 2–4 versus scaled time $\tau = \lambda t$. The left plot for $\gamma_1 = \gamma_1 = 0$ and the right plot for $\gamma_1 = \gamma_1 = 0.0005$. In Figure 2a,b, we have considered the time-dependent coupling parameter $\mu/\lambda = 0$ in the absence of the detuning parameters and Kerr-medium ($\Delta_1/\lambda = \Delta_2/\lambda = \chi_1 = \chi_2 = 0$). The behavior of the atomic population inversion in Figure 2a exhibits the collapse and revival phenomena. The number of oscillations in Figure 2b is less than that in Figure 2a. Also, in Figure 2b, the effect of the photon assisted atomic phase damping parameter leads to decreasing the amplitude of oscillations as time develops and the mean value of oscillations become zero in the time evolution process. In Figure 2c,d in which the value of the time-dependent coupling parameter $\mu/\lambda = 3Pi$, the behavior of the atomic population inversion in Figure 2c is changed compared with Figure 2a. The collapse intervals is elongated. In Figure 2e, when $\mu/\lambda = 10Pi$, we observe that the behavior of the atomic population inversion has started only with a short period of revivals followed by a long time-interval of collapse compared with the previous cases. This means that we can consider the time-dependent coupling as a quantum control parameter. The effect of the detunning parameters on the atomic population inversion in the absence or presence of the photon assisted atomic phase damping parameter and in the absence of both of the time-dependent coupling parameter and Kerr medium ($\mu/\lambda = \chi_1 = \chi_2 = 0$) appeared in Figure 3. In Figure 3a, when $\Delta_1/\lambda = \Delta_2/\lambda = 7$, $\gamma_1 = \gamma_1 = 0$, we have along intervals of collapses compared with that in Figure 2a. In addition, in Figure 3b, the effect of the photon-assisted atomic phase damping parameter leads to decreasing the amplitude of oscillations as time develops, and the mean value of oscillations becomes zero in the time evolution process. By the increase of the value of the detuning parameter, the collapse interval is elongated, so we can con consider the detuning parameter as a quantum control parameter (see Figure 3c–f). The behavior of the atomic population inversion in Figure 3e,f is similar to that in Figure 2e,f ($\Delta_1/\lambda = \Delta_3/\lambda = 25$). To discuss the influence of the Kerr medium on the atomic population inversion in the absence or presence of the photon-assisted atomic phase damping parameter as well as in the absence of both of the time-dependent coupling parameter and detuning parameters ($\mu/\lambda = \Delta_1/\lambda = \Delta_2/\lambda = 0$), we have plotted Figure 4. For a small value of Kerr medium parameter ($\chi_1 = \chi_2 = 0.01$), the behavior of $W(\tau)$ in Figure 4a is changed compared to the behavior of $W(\tau)$ in Figures 2a and 3a; the amplitude of oscillations is decreased. By the increase of the value of Kerr medium, the behavior of $W(\tau)$ changes. The mean value of oscillations is shifted upward. For a great value of Kerr medium, the behavior of the atomic population inversion is completely changed compared with the previous cases. We have the greatest negative mean value of oscillations and the maximum value of fluctuations approaches one (see Figure 4e). This means that the energy increases in the atomic system. The photon-assisted atomic phase damping parameter leads to destroying the amplitude of oscillations as time develops (see Figure 4b,d,f).

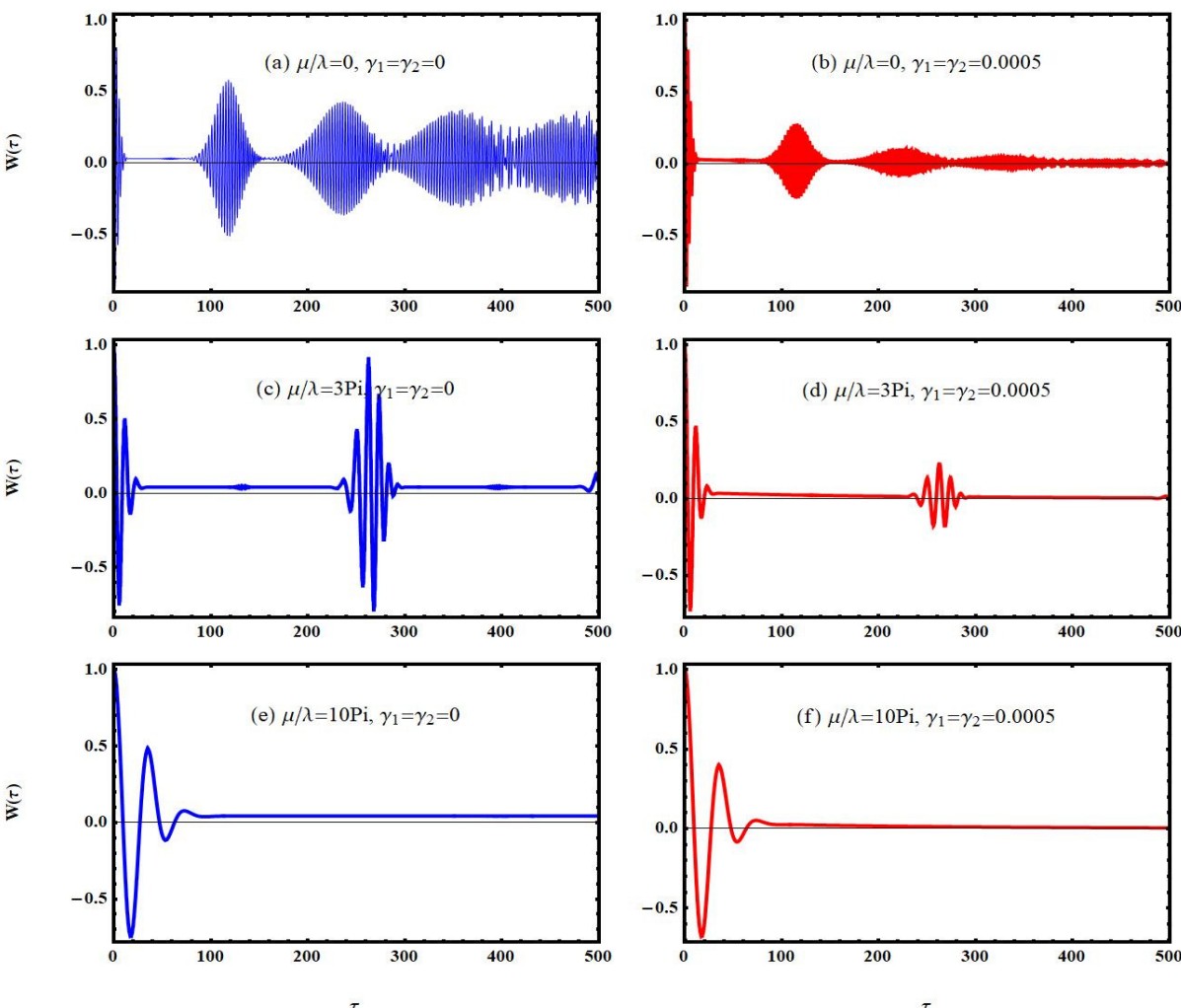

**Figure 2.** Evolution of the atomic population inversion $W(\tau)$ of a three-level $\Xi$-type atom interacting with a two-mode coherent field for the parameters $\bar{n}_1 = \bar{n}_2 = 10$, $\Delta_1 = \Delta_2 = 0$, $\chi_1 = \chi_2 = 0$ and for: (**a**) $\mu/\lambda = 0$, $\gamma_1 = \gamma_2 = 0$, (**b**) $\mu/\lambda = 0$, $\gamma_1 = \gamma_2 = 0.0005$, (**c**) $\mu/\lambda = 3Pi$, $\gamma_1 = \gamma_2 = 0$, (**d**) $\mu/\lambda = 3Pi$, $\gamma_1 = \gamma_2 = 0.0005$, (**e**) $\mu/\lambda = 10Pi$, $\gamma_1 = \gamma_2 = 0$, (**f**) $\mu/\lambda = 10Pi$, $\gamma_1 = \gamma_2 = 0.0005$.

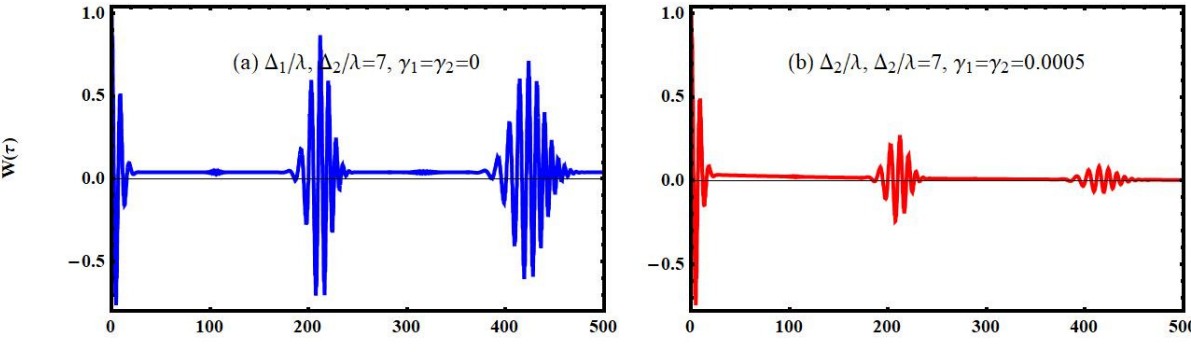

**Figure 3.** *Cont.*

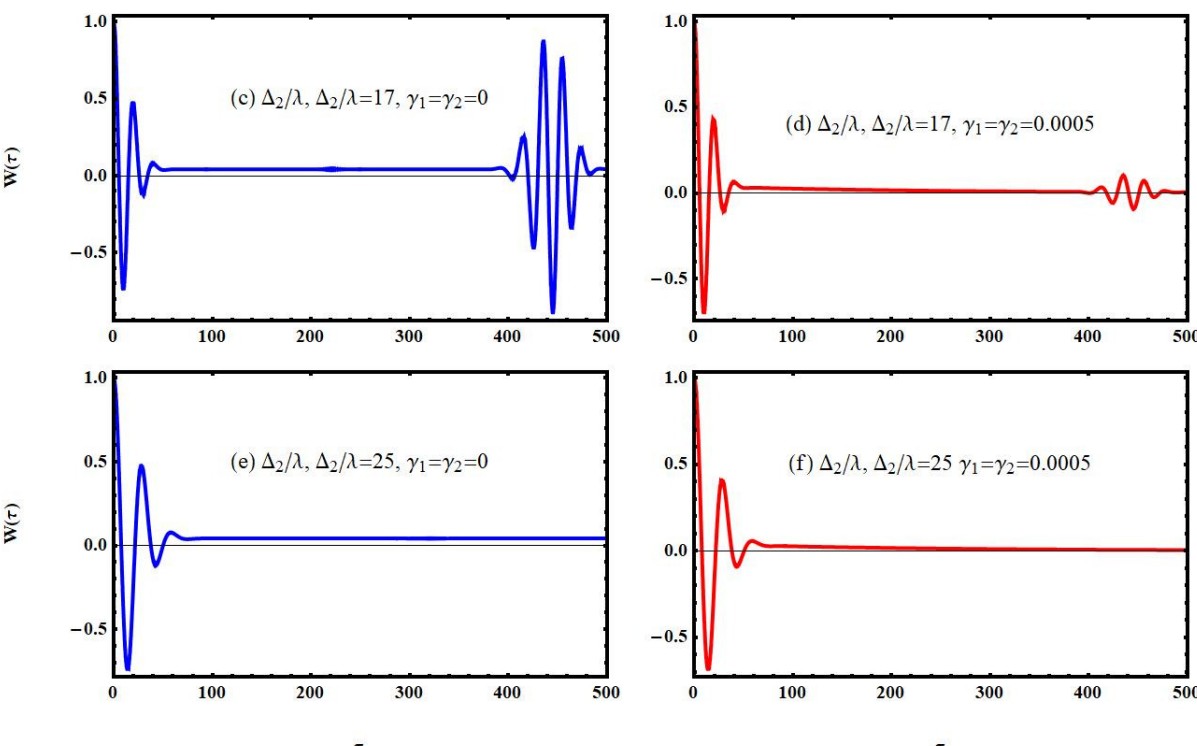

**Figure 3.** The same as in Figure 2 but for the parameters $\overline{n}_1 = \overline{n}_2 = 10$, $\mu/\lambda = 0$, $\chi_1 = \chi_2 = 0$ and for: (**a**) $\Delta_1/\lambda = \Delta_2/\lambda = 7$, $\gamma_1 = \gamma_2 = 0$, (**b**) $\Delta_1/\lambda = \Delta_2/\lambda = 7$, $\gamma_1 = \gamma_2 = 0.0005$, (**c**) $\Delta_1/\lambda = \Delta_2/\lambda = 17$, $\gamma_1 = \gamma_2 = 0$, (**d**) $\Delta_1/\lambda = \Delta_2/\lambda = 17$, $\gamma_1 = \gamma_2 = 0.0005$, (**e**) $\Delta_1/\lambda = \Delta_2/\lambda = 25$, $\gamma_1 = \gamma_2 = 0$, (**f**) $\Delta_1/\lambda = \Delta_2/\lambda = 25$, $\gamma_1 = \gamma_2 = 0.0005$.

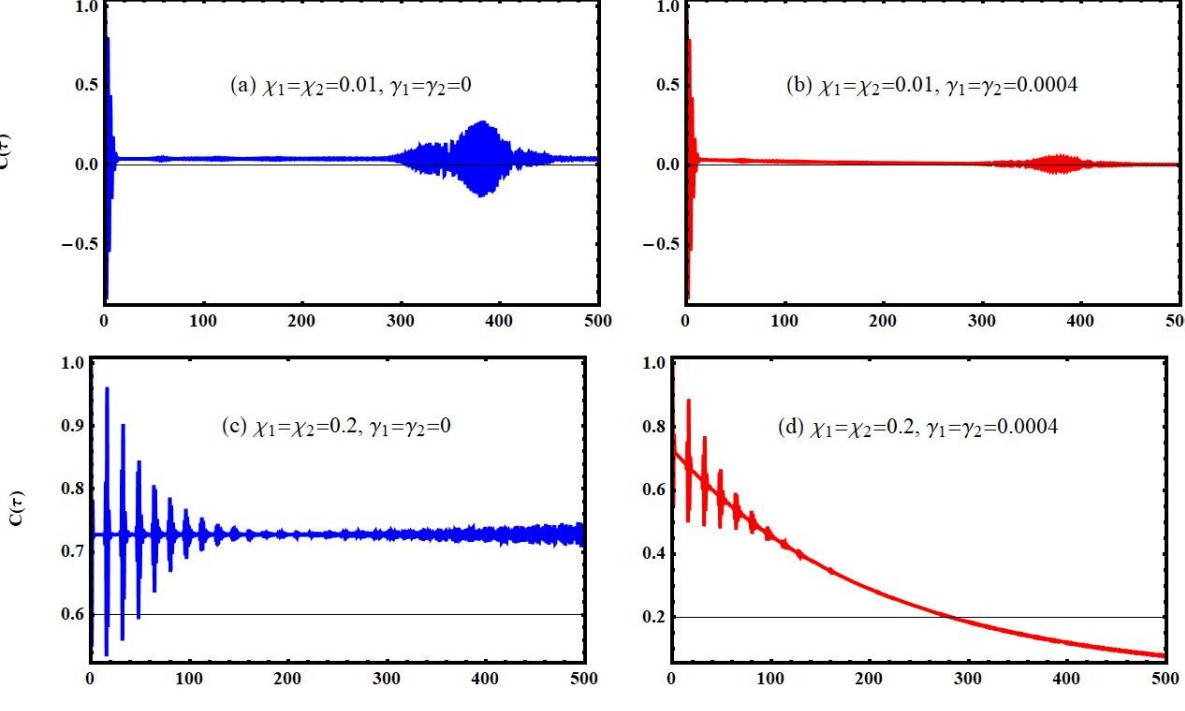

**Figure 4.** *Cont.*

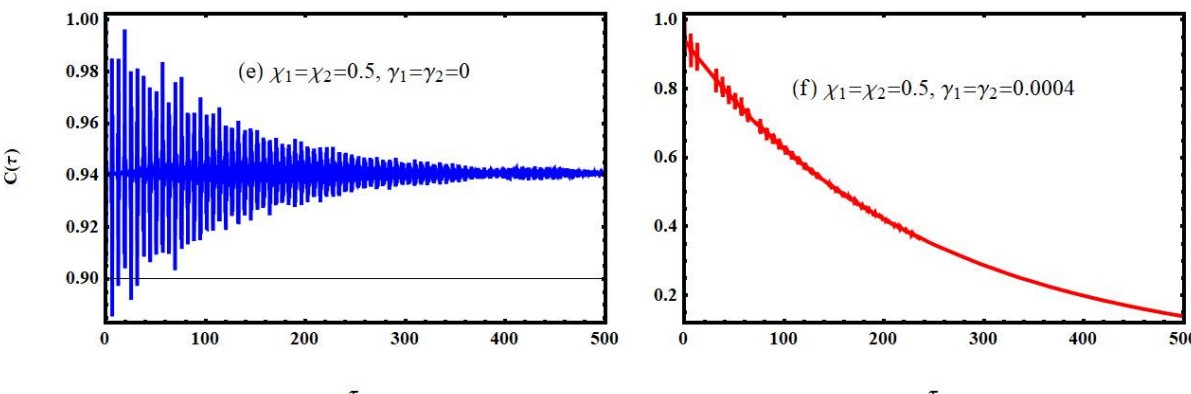

**Figure 4.** The same as in Figure 2 but for the parameters $\bar{n}_1 = \bar{n}_2 = 10$, $\mu/\lambda = 0$, $\Delta_1 = \Delta_2 = 0$ and for: (**a**) $\chi_1 = \chi_2 = 0.01$, $\gamma_1 = \gamma_2 = 0$, (**b**) $\chi_1 = \chi_2 = 0.01$, $\gamma_1 = \gamma_2 = 0.0004$, (**c**) $\chi_1 = \chi_2 = 0.2$, $\gamma_1 = \gamma_2 = 0$, (**d**) $\chi_1 = \chi_2 = 0.2$, $\gamma_1 = \gamma_2 = 0.0004$, (**e**) $\chi_1 = \chi_2 = 0.5$, $\gamma_1 = \gamma_2 = 0$, (**f**) $\chi_1 = \chi_2 = 0.5$, $\gamma_1 = \gamma_2 = 0.0004$.

## 4. Concurrence

The concurrence is presented by Wootters and Hill [34,35] as a proper measure of the entanglement of any state of two qubits, pure or mixed. For a pure state $|\Psi(t)\rangle$ on $(K \times L)$-dimensional Hilbert space $M = M_K \otimes M_L$, the concurrence can be defined as follows [14,36].

$$C(t) = \sqrt{2[|\langle\Psi(t)|\Psi(t)\rangle|^2 - Tr(\varrho_L^2(t))]},$$

where $\varrho_L(t) = Tr_K(|\Psi(t)\rangle\langle\Psi(t)|)$ is the reduced density operator of the subsystem with dimension $L$ and $Tr_K$ is the partial trace over $M_K$. It is remarkable to mention that the concurrence fluctuates between $\sqrt{2(L-1)}$ for a maximally entangled state and 0 for a separable state. Herein, we calculate the concurrence to obtain the degree of entanglement (DEM) between the atom and the field. Using Equation (14), we can rewrite concurrence in the following form

$$C(t) = \sqrt{2 \sum_{i,j-1,2,...9}^{i\neq j} [\varrho_{ii}(t)\varrho_{jj}(t) - \varrho_{ij}(t)\varrho_{ji}(t)]}.$$

Now, we are going to study the evolution of the concurrence $C(\tau)$ versus the scaled time $\tau = \lambda t$ for the same parameters that we used in Figures 2–4. An illustration of the time evolution of the concurrence for $\gamma_1 = \gamma_1 = 0$ (left plot), $\gamma_1 = \gamma_1 = 0.001$ (right plot) and for $\bar{n}_1 = \bar{n}_2 = 10$ is shown in Figures 5–7. In Figure 5a, when all parameters are zero, we note that $C(\tau)$ starts from zero; then, it is followed by a sequence of fluctuations in the oscillation. This means that this system begins in a disentangled state; then, it develops into a mixed state ($t > 0$). It is clear that in Figure 5c,e, the time-dependent coupling parameter plays a dramatic role in the degree of entanglement: the maximum value of $C(\tau)$ decreased and the periodic behavior appeared. In addition, the time interval of the period is elongated as the value of $\mu/\lambda$ increases. In Figure 5b,d,f, we observed that the photon-assisted atomic phase damping parameter ($\gamma_1 = \gamma_1 = 0.001$) leads to a decrease in the degree of entanglement between the atom and the field and finally vanishes as the time develops (i.e., no entanglement).

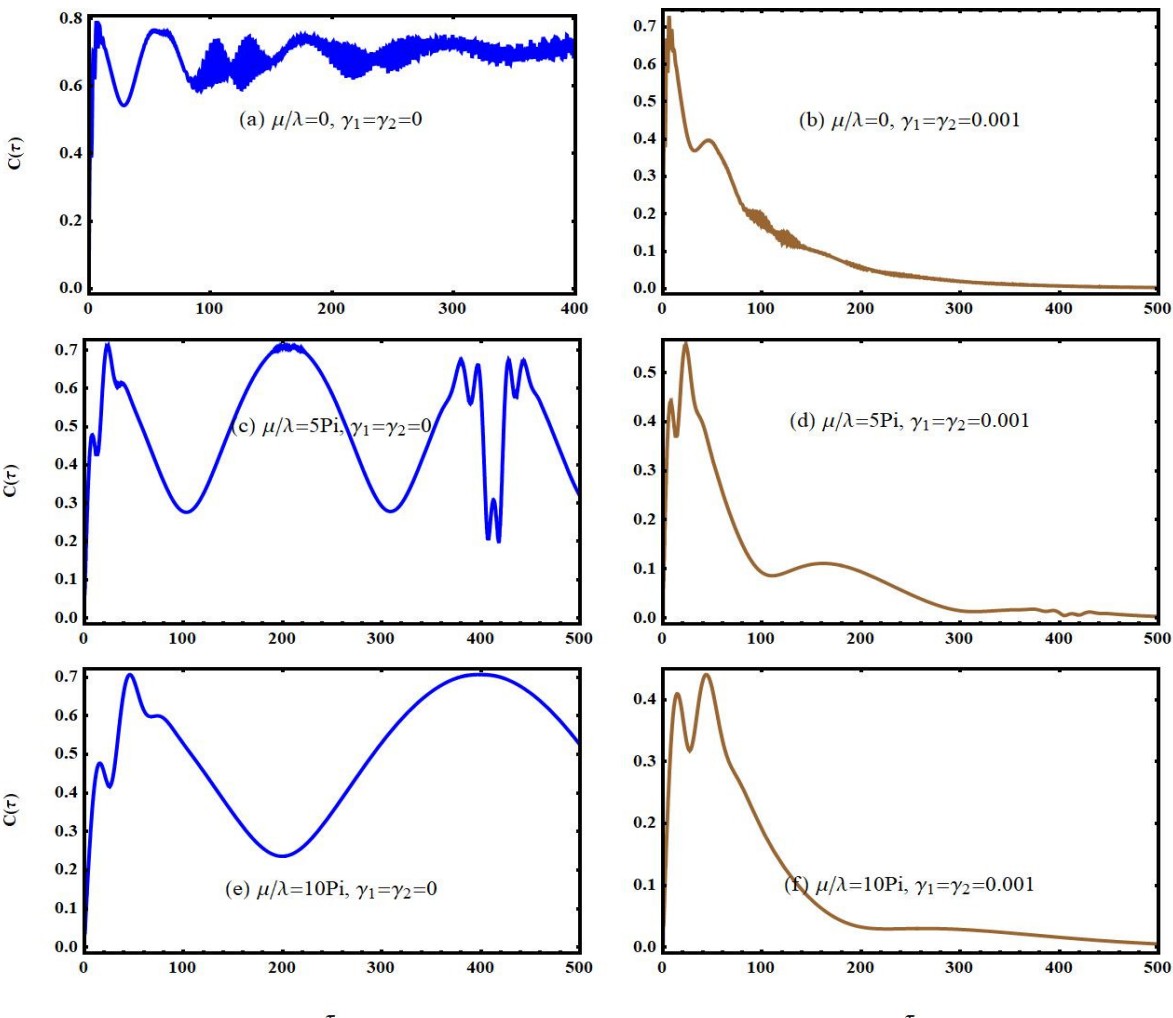

**Figure 5.** Evolution of the concurrence $C(\tau)$ of a three-level $\Xi$-type atom interacting with a two-mode coherent field for the parameters $\bar{n}_1 = \bar{n}_2 = 10$, $\Delta_1 = \Delta_2 = 0$, $\chi_1 = \chi_2 = 0$ and for: (**a**) $\mu/\lambda = 0$, $\gamma_1 = \gamma_2 = 0$, (**b**) $\mu/\lambda = 0$, $\gamma_1 = \gamma_2 = 0.001$, (**c**) $\mu/\lambda = 5Pi$, $\gamma_1 = \gamma_2 = 0$, (**d**) $\mu/\lambda = 5Pi$, $\gamma_1 = \gamma_2 = 0.001$, (**e**) $\mu/\lambda = 10Pi$, $\gamma_1 = \gamma_2 = 0$, (**f**) $\mu/\lambda = 10Pi$, $\gamma_1 = \gamma_2 = 0.001$.

To explore the effect of the detuning parametes $\Delta_1/\lambda$, $\Delta_2/\lambda$ on $C(\tau)$ in the absence or presence of the photon-assisted atomic phase damping parameter and in the absence of both of the time-dependent coupling parameter and Kerr medium ($\mu/\lambda = \chi_1 = \chi_2 = 0$), we have plotted Figure 6. In Figure 6a, when $\Delta_1/\lambda = \Delta_2/\lambda = 10$, $\gamma_1 = \gamma_1 = 0$, we observed that the maximum value of $C(\tau)$ decreased compared with Figure 5a. In addition, the periodic behavior appeared in Figure 6a. As the detuning parameter increases, the time interval of the period is elongated (see Figure 6c,e). The effect of the detuning parameter in the presence of the photon-assisted atomic phase damping parameter leads to a decrease in the degree of entanglement between the atom and the field as the time develops (see Figure 6b,d,f). To visualize the influence of the Kerr medium on the concurrence $C(\tau)$ in the absence or presence of the photon-assisted atomic phase damping parameter and in the absence of both of the time-dependent coupling parameter and detuning parameters ($\mu/\lambda = \Delta_1/\lambda = \Delta_2/\lambda = 0$), we have plotted Figure 7. We notice that when $\chi_1/\lambda = \chi_2/\lambda = 0.01$, $\gamma_1 = \gamma_1 = 0$, the nonlinear interaction of the Kerr medium with the field modes leads to increasing the maximum value of the concurrence with the decreasing of the amplitude of oscillations (see Figure 7a). By the increase of the nonlinear interaction of the Kerr medium with field modes, the maximum value of the concurrence decreases, and then, the degree of entanglement between the atom and the field decreases (see Figure 7c,e). In addition, we observed that when $\chi_1/\lambda = \chi_2/\lambda = 0.5$, $\gamma_1 = \gamma_1 = 0$,

many oscillations have appeared. The effect of Kerr medium in the presence of the photon-assisted atomic phase damping parameter leads to a decrease in the degree of entanglement between the atom and the field in the time evolution process (see Figure 7b,d,f).

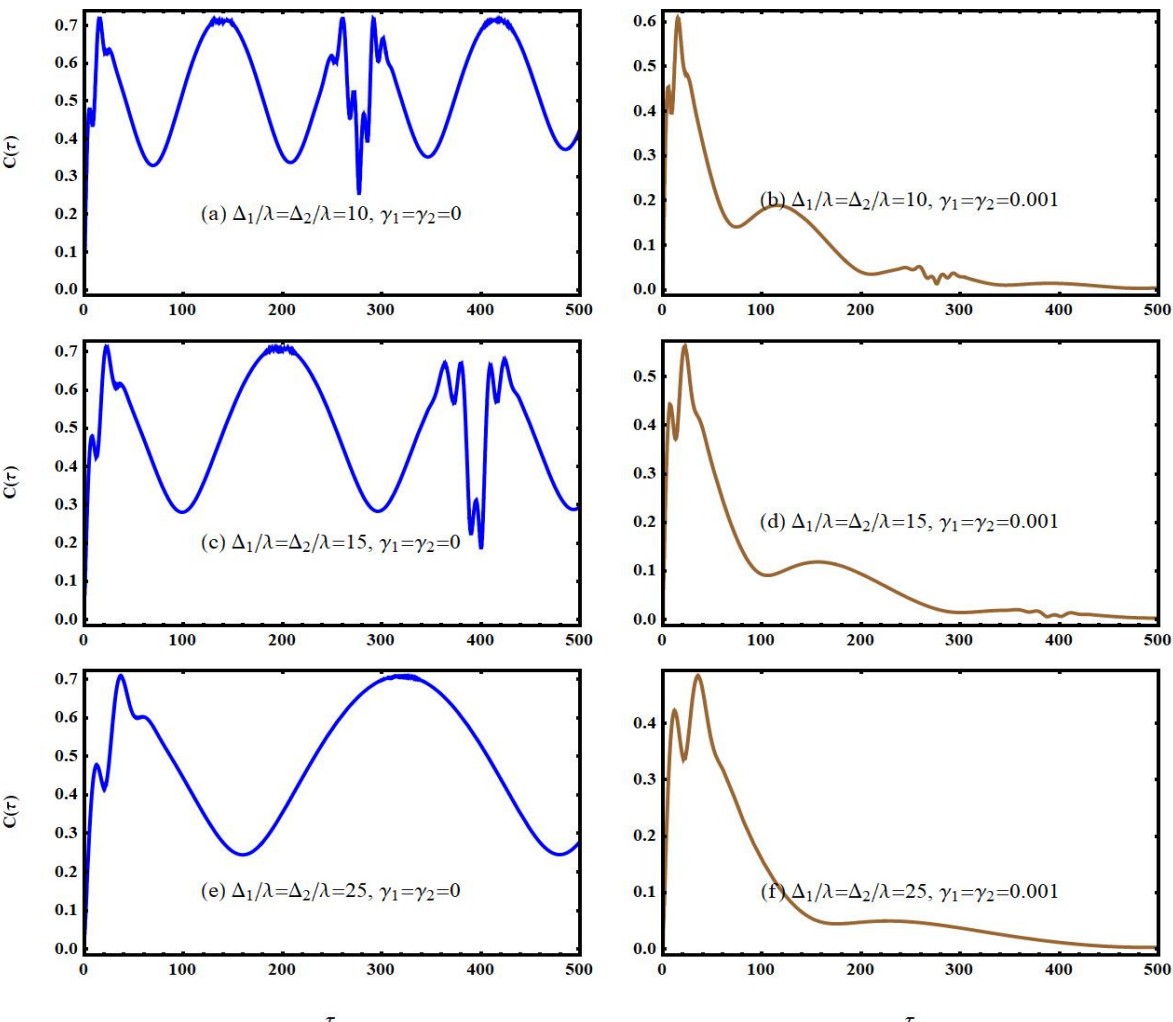

**Figure 6.** The same as in Figure 5 but for the parameters $\bar{n}_1 = \bar{n}_2 = 10$, $\mu/\lambda = 0$, $\chi_1 = \chi_2 = 0$ and for: (**a**) $\Delta_1/\lambda = \Delta_2/\lambda = 10$, $\gamma_1 = \gamma_2 = 0$, (**b**) $\Delta_1/\lambda = \Delta_2/\lambda = 10$, $\gamma_1 = \gamma_2 = 0.001$, (**c**) $\Delta_1/\lambda = \Delta_2/\lambda = 15$, $\gamma_1 = \gamma_2 = 0$, (**d**) $\Delta_1/\lambda = \Delta_2/\lambda = 15$, $\gamma_1 = \gamma_2 = 0.001$, (**e**) $\Delta_1/\lambda = \Delta_2/\lambda = 25$, $\gamma_1 = \gamma_2 = 0$, (**f**) $\Delta_1/\lambda = \Delta_2/\lambda = 25$, $\gamma_1 = \gamma_2 = 0.001$.

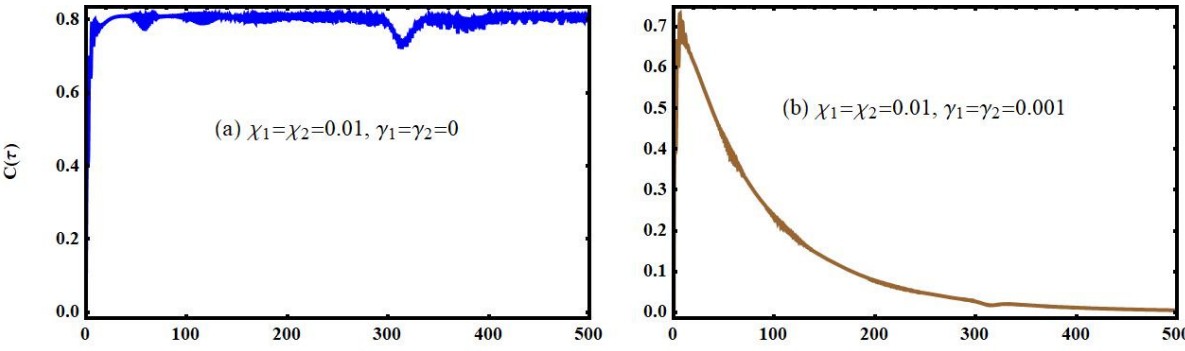

**Figure 7.** *Cont.*

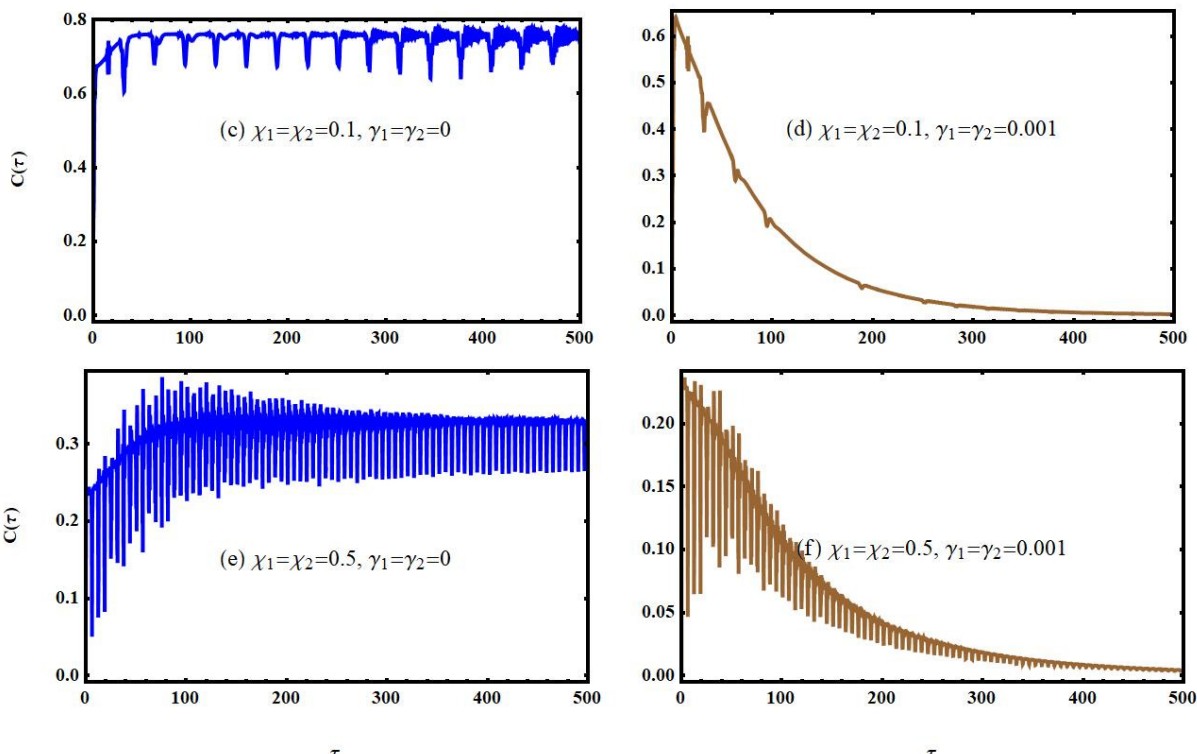

**Figure 7.** The same as in Figure 5 but for the parameters $\bar{n}_1 = \bar{n}_2 = 10$, $\mu/\lambda = 0$, $\Delta_1 = \Delta_2 = 0$ and for: (**a**) $\chi_1 = \chi_2 = 0.01$, $\gamma_1 = \gamma_2 = 0$, (**b**) $\chi_1 = \chi_2 = 0.01$, $\gamma_1 = \gamma_2 = 0.001$, (**c**) $\chi_1 = \chi_2 = 0.1$, $\gamma_1 = \gamma_2 = 0$, (**d**) $\chi_1 = \chi_2 = 0.1$, $\gamma_1 = \gamma_2 = 0.001$, (**e**) $\chi_1 = \chi_2 = 0.5$, $\gamma_1 = \gamma_2 = 0$, (**f**) $\chi_1 = \chi_2 = 0.5$, $\gamma_1 = \gamma_2 = 0.001$.

## 5. Conclusions

In summary, we have examined the interaction of a three-level $\Xi$-type atom with a two-mode field taking into account new parameters such as the photon-assisted atomic phase damping parameter, Kerr medium and the detuning parameter. Furthermore, the coupling parameter is modulated to be time-dependent. Under the Rotating-Wave Approximation (RWA), the exact solution of the model is obtained for the state vector of the whole system. The influence of the photon-assisted atomic phase damping parameter, the time-dependent coupling parameter, detuning parameter and Kerr nonlinearity on the atomic population inversion and the concurrence have been studied. It is shown that the atomic population inversion has the quantum collapse–revival behavior, and the time-dependent coupling parameter and detuning parameter can be considered as quantum control parameters. The concurrence of a three-level atomic system has been introduced, and its time evolution has been studied, providing the ability to explore the degree of entanglement of the available systems in the absence or presence of the photon-assisted atomic phase damping parameter. Finally, we can deduce that the presence of the time-dependent coupling parameter, detuning parameter, Kerr nonlinearity and the photon-assisted atomic phase damping parameter leads to noticeable effects in the quantum entanglement of the considered systems.

**Author Contributions:** Methodology, S.K.; Supervision, M.A.-A. All authors have read and agreed to the published version of the manuscript.

**Funding:** This research received no external funding.

**Data Availability Statement:** Not applicable.

**Conflicts of Interest:** The authors declare no conflict of interest.

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
