# Peer review of "Quantum Control of a Nonlinear Time-Dependent Interaction of a Damped Three-Level Atom"

_axioms, doi:10.3390/axioms12060552_

Round 1
Reviewer 1 Report
Quantum Control of a Nonlinear Time-Dependent Interaction of a Damped Three-Level Atom
Authors discussed the non-linear interaction between a three-level atom and a bimodal field.
1) First of all, this manuscript possesses lots of typos. Appropriate spell checks are required indeed. English language and style are not OK.
2) All discussions of this manuscript are nothing new. They basically repeated well-known results available in literature, without delivering additional valuable new messages.
I would not like to criticize this manuscript without any scientific justification. However, this manuscript is not ready for publication in terms of the scientific depth.
Author Response
1-Typos have been corrected.
2-Discussions have been updated.
3-Latex does not recognize some physical words (such as detuning,...), so they appear as errors.
Reviewer 2 Report
The paper deals with an important problem in quantum optics, namely the behavior of a three-level atom in a multi-mode electromagnetic field. Tha applications directly relate to quantum information science, and the authors advanced this field in their new paper. Their results are well justified mathematically and supported by numerical analysis.
My only concern is excessive self-citation by one the authors. Out of 41 references, 16 are co-written by the author, Mahmoud Abdel-Aty. I recommend revising the reference list, and retaining only most essential references. Once this issue is addressed, I would be glad to recommend the manuscript for publication.
Author Response
1-The list of references has been revised, and only the most important references have been retained.
2-Typos have been corrected.
Reviewer 3 Report
On page 2 the authors should include a few sentences on what is new in their work in comparison of the related literature that they have sited.
On page 3 there is a typo on the line after Eq. (1), I think the subscript should be ij rather than i. On the next line the operator b should also be mentioned.
Author Response
1-On page 2, we have included a few sentences on what is new in our work in comparison of the related literature that we have sited.
2-On page 3, the typo on the line after Eq. (1) has been modified to ij. Also, On the next line the operator b has been mentioned.
Round 2
Reviewer 1 Report
The present version may be published.